# Three Particle Muon-Electron Bound Systems in Quantum Electrodynamics

Alexey V. Eskin [1,†] , Vladimir I. Korobov [2,†] , Alexei P. Martynenko [1,*,†] and Fedor A. Martynenko [1,†]

[1] Samara U., Moskovskoe Shosse 34, 443086 Samara, Russia
[2] BLTP JINR, 141980 Dubna, Russia
[*] Correspondence: a.p.martynenko@samsu.ru
[†] These authors contributed equally to this work.

**Abstract:** The muonic 2P-2S Lamb shift in muon-electron atoms and ions of helium, lithium, beryllium, and boron with the electron in the ground state was calculated by the perturbation theory using the fine structure constant and the electron-muon mass ratio. The corrections of first- and second-orders of perturbation theory on the Coulomb interaction and nucleus recoil were taken into account. The obtained analytical results were validated numerically using calculations within the variational method.

**Keywords:** muonic atoms and ions; quantum electrodynamics; fine structure





## 1. Introduction

The study of the energy levels of three-particle Coulomb systems is one of the fundamental problems of atomic physics that has practical applications in the study of muon catalysis reactions. The energy spectrum of three-particle muon-electron-nucleus systems has been studied in a number of experiments along with two-particle muonic atoms [1–3]. New plans for precision microwave spectroscopy of the J-PARC MUSE [4] collaboration involved measuring the hyperfine structure (HFS) of the ground state of muonic helium with an accuracy two orders of magnitude higher than the accuracy of previous experiments in the 1980s [5].

Usually, when calculating the fine and hyperfine structures of the spectrum in muon-electron-nucleus systems, several methods have been used. A number of approaches use the adiabatic representation of the three-body problem, the variational method, which allows one to find wave functions and energies with very high accuracy [6–15]. Another analytical method for calculating the energy levels of such three-particle systems was formulated in [16,17] and applied to calculate the hyperfine structure of the spectrum and the electron Lamb shift in [18–22]. It was based on the use of the perturbation theory (PT) method, according to the following parameters: the fine structure constant $\alpha$ and the mass ratio of the electron and muon.

In the last decade, the CREMA collaboration has measured the Lamb shift in a number of the simplest muonic atoms [1,2] and obtained more precise values of the charge radius of the light nuclei. If there was a significant discrepancy in the results obtained for the charge radii of the proton and deuteron of muonic hydrogen and muonic deuterium, as compared to the previous values found in studies of electronic systems, then such a discrepancy was not found for muonic helium. In [2], the experimental determination of the Lamb shift and the fine structure of the muonic helium ion was carried out using a previously developed measurement procedure for muonic hydrogen and deuterium. In the experiment, approximately 500 negative muons per second with ultra-low energy (a few keV) were stopped in helium gas at a low pressure of 2 mbar and room temperature. As a result of these collisions, the muon ejected an electron and was captured by the He atom, forming a three-particle muon-electron atom in a highly excited state (with the

principal quantum number $n = 14$). For orbits with a high principal quantum number n, the Auger rates were much higher than the rates of the radiative transitions. As a result, the remaining electron was ejected from the atom and a helium muonic ion was formed. Within approximately 100 ns, it transitioned to the $1S$ ground state or to the $2S$ metastable state via radiative transitions. Approximately one percent of muons populated the $2S$ $(\mu He - 4)^+$ metastable state, which has a lifetime of 1.75 μs. When a low helium gas pressure of 2 mbar was used, the helium ions were not neutralized. At the same time, if some individual atoms of neutral muonic helium were present, then the bound electron had little effect on the measurement of the muon Lamb shift. A precision calculation of the energy spectrum of two-particle muonic ions was performed in [23,24]. The purpose of this work was to take into account the effects of the presence of an electron in a muonic helium atom when measuring the muon Lamb shift. We expanded on [21,22], which studied the energy levels of muon-electron atoms and ions of helium, lithium, beryllium, and boron, as related to the muon Lamb shift in the framework of the analytical perturbation theory (PT) and the variational method. We investigated how the presence of an electron affected the measurement of the muon Lamb shift and the charge radius of the nucleus.

## 2. General Formalism

To calculate the energy levels by analytical perturbation theory, we divided the Hamiltonian of the system into several parts, including the main contribution of the Coulomb interaction in the term $H_0$ in the form:

$$H = H_0 + \Delta H + \Delta H_{rec} + \Delta H_{vp} + \Delta H_{str} + \Delta H_{vert}, \tag{1}$$

$$H_0 = -\frac{1}{2M_\mu}\nabla_\mu^2 - \frac{1}{2M_e}\nabla_e^2 - \frac{Z\alpha}{x_\mu} - \frac{(Z-1)\alpha}{x_e}, \tag{2}$$

$$\Delta H = \frac{\alpha}{|\mathbf{x}_\mu - \mathbf{x}_e|} - \frac{\alpha}{x_e}, \quad \Delta H_{rec} = -\frac{1}{M}\nabla_\mu \cdot \nabla_e, M_e = \frac{m_e M}{(m_e + M)}, \quad M_\mu = \frac{m_\mu M}{(m_\mu + M)}, \tag{3}$$

where $\mathbf{x}_\mu$ and $\mathbf{x}_e$ are the muon and electron radius vectors with respect to the nucleus, and $Ze$ is the nuclear charge. The terms $\Delta H_{vp}$, $\Delta H_{str}$, and $\Delta H_{vert}$ denote the contributions to vacuum polarization, nuclear structure, and vertex corrections.

In the initial approximation, which was determined by the Hamiltonian $H_0$, the wave function of the system has a simple analytical form:

$$\Psi_{2S}(\mathbf{x}_e, \mathbf{x}_\mu) = \psi_{e\,1S}(\mathbf{x_e})\psi_{\mu\,2S}(\mathbf{x}_\mu) = \frac{(W_e W_\mu)^{3/2}}{2\sqrt{2\pi}}\left(1 - \frac{W_\mu x_\mu}{2}\right)e^{-\frac{W_\mu x_\mu}{2}}e^{-W_e x_e}, \tag{4}$$

$$\Psi_{2P}(\mathbf{x}_e, \mathbf{x}_\mu) = \psi_{e\,1S}(\mathbf{x_e})\psi_{\mu\,2P}(\mathbf{x}_\mu) = \frac{(W_e W_\mu)^{3/2}}{2\sqrt{6}}(W_\mu x_\mu)e^{-\frac{W_\mu x_\mu}{2}}e^{-W_e x_e}(\boldsymbol{\varepsilon}\mathbf{n}), \tag{5}$$

$$W_\mu = Z\alpha M_\mu, \quad W_e = (Z-1)\alpha M_e, \tag{6}$$

which makes it possible to calculate the corrections using the perturbation theory. In (5), the angular part of the wave function was represented in tensor form, $\boldsymbol{\varepsilon}$ is the polarization vector, and $\mathbf{n} = \mathbf{x}_\mu / x_\mu$.

In the initial approximation, the energy of a bound system was determined by the sum of the Coulomb energies of an electron and a muon:

$$E_{2S,2P} = -\frac{1}{2}M_e(Z-1)^2\alpha^2 - \frac{1}{8}M_\mu Z^2\alpha^2. \tag{7}$$

In the energy shift $(2P - 2S)$, these contributions canceled out. In the first order PT, the Coulomb interaction $\Delta H$ resulted in shifts that were determined by matrix elements:

$$\Delta E^{(1)}(2S) = \left\langle \Psi_{2S} \left| \left( \frac{\alpha}{x_{\mu e}} - \frac{\alpha}{x_e} \right) \right| \Psi_{2S} \right\rangle = W_e \alpha \left( -28a_1^2 + 220a_1^3 - 1152a_1^4 \ldots \right),$$

$$\Delta E^{(1)}(2P) = \left\langle \Psi_{2P} \left| \left( \frac{\alpha}{x_{\mu e}} - \frac{\alpha}{x_e} \right) \right| \Psi_{2P} \right\rangle = W_e \alpha \left( -20a_1^2 + 140a_1^3 - 672a_1^4 \ldots \right),$$

(8)

$$\Delta E^{(1)}(2S - 2P) = W_e \alpha \frac{8a_1^2}{(1 + 2a_1)^5}, \quad a_1 = \frac{W_e}{W_\mu}, \tag{9}$$

where $x_{\mu e} = |\mathbf{x}_\mu - \mathbf{x}_e|$. The results of (8) are presented as an expansion of the small parameter $a_1$.

### 2.1. The Coulomb Corrections in Second Order PT

The general expression for the correction to the energy levels in the second order of the perturbation theory (PT), with respect to the $\Delta H$ interaction when the muon was in the intermediate state $n = 2S$, had the following form:

$$\Delta E_{2S}^{(2)}(n = 2S) = \int \psi_{\mu 2S}(\mathbf{x}_\mu) \psi_{e1S}(\mathbf{x}_e) \left( \frac{\alpha}{x_{\mu e}} - \frac{\alpha}{x_e} \right) \psi_{\mu 2S}(\mathbf{x}_\mu) \left( \frac{\alpha}{x'_{\mu e}} - \frac{\alpha}{x'_e} \right) \times \tag{10}$$

$$\psi_{\mu 2S}(\mathbf{x'}_\mu) \psi_{e1S}(\mathbf{x'}_e) \tilde{G}_{e1S}(\mathbf{x}_e, \mathbf{x'}_e) \psi_{\mu 2S}(\mathbf{x'}_\mu) d\mathbf{x}_\mu d\mathbf{x'}_\mu d\mathbf{x}_e d\mathbf{x'}_e,$$

where $\tilde{G}_{e1S}(\mathbf{x}_e, \mathbf{x'}_e)$ is the reduced Coulomb Green's function of the electron for the $1S$ state [25]:

$$\tilde{G}_{e1S}(\mathbf{x}_1, \mathbf{x}_3) = \sum_{n \neq 1S}^{\infty} \frac{\psi_{en}(\mathbf{x}_3) \psi_{en}^*(\mathbf{x}_1)}{E_{e1S} - E_{en}} = -\frac{W_e M_e}{\pi} e^{-W_e(x_1 + x_3)} \left[ \frac{1}{2W_e x_>} - \right. \tag{11}$$

$$\left. - \ln(2W_e x_>) - \ln(2W_e x_<) + Ei(2W_e x_<) + \frac{7}{2} - 2C - W_e(x_1 + x_3) + \frac{1 - e^{2W_e x_<}}{2W_e x_<} \right],$$

where $x_< = \min(x_1, x_3)$, $x_> = \max(x_1, x_3)$, $C = 0.577216\ldots$ is the Euler constant, and $Ei(x)$ is integral exponential function.

The expression (10) contains two identical integrals, which were calculated analytically:

$$V_\mu(\mathbf{x}_e) = \int d(\mathbf{x}_\mu) |\psi_{\mu 2S}(\mathbf{x}_\mu)|^2 \left( \frac{\alpha}{x_{\mu e}} - \frac{\alpha}{x_e} \right) = -\frac{\alpha e^{-W_\mu x_e}}{8x_e} \left( 8 + 6W_\mu x_e + 2(W_\mu x_e)^2 + (W_\mu x_e)^3 \right). \tag{12}$$

After analytical integration and expansion in $a_1$ and using (12) and the explicit form $\tilde{G}_{e1S}(\mathbf{x}_e, \mathbf{x'}_e)$ [21,23,25], we obtained the following:

$$\Delta E_{2S}^{(2)}(n = 2S) = -M_e \alpha^2 a_1^3 \left[ \frac{5993}{64} + a_1 \left( -\frac{24111}{64} - 784 \ln 4a_1 \right) + \ldots \right]. \tag{13}$$

A similar result for a muon in the $2P$ state had the form:

$$\Delta E_{2P}^{(2)}(n = 2P) = -M_e \alpha^2 a_1^3 \left[ \frac{31329}{576} + a_1 \left( -\frac{26965}{192} - 400 \ln 4a_1 \right) + \ldots \right]. \tag{14}$$

Now, we assumed the muon was in an intermediate state that would not coincide with $2S$ (or $2P$). In the second order PT, such a contribution was determined by the following expression:

$$\Delta E_{2S}^{(2)}(n \neq 2S) = \int \psi_{\mu 2S}(\mathbf{x}_\mu) \psi_{e1S}(\mathbf{x}_e) \left( \frac{\alpha}{x_{\mu e}} - \frac{\alpha}{x_e} \right) \psi_{\mu 2S}(\mathbf{x'}_\mu) d\mathbf{x}_e d\mathbf{x'}_e d\mathbf{x}_\mu d\mathbf{x'}_\mu \tag{15}$$

$$\times \sum_{n\neq 2S} \psi_{\mu n}(\mathbf{x}_\mu)\psi_{\mu n}^*(\mathbf{x}'_\mu) \sum_{n'} \frac{\psi_{en'}(\mathbf{x}_e)\psi_{en'}^*(\mathbf{x}'_e)}{E_{e1S} - E_{en'} + E_{\mu 2S} - E_{\mu n}} \left(\frac{\alpha}{x'_{\mu e}} - \frac{\alpha}{x'_e}\right)\psi_{e1S}(\mathbf{x}'_e).$$

Equation (15) contains the Coulomb Green's function of the electron, which we replaced in the leading order in $a_1$ with the free Green's function:

$$G_e(\mathbf{x}_e, \mathbf{x}'_e) \approx -\frac{M_e\alpha^2}{2\pi}\frac{e^{-b_n|\mathbf{x}_e - \mathbf{x}'_e|}}{|\mathbf{x}_e - \mathbf{x}'_e|}, \quad b_n = \sqrt{2M_e(E_{\mu n} - E_{\mu 2S} - E_{e1S})}. \tag{16}$$

Note that this approximation could be improved by using a PT series for the electron Green's function, as in [22]. Integration over coordinates in (15) could be achieved using the completeness condition:

$$\sum_{n\neq 2S} \psi_{\mu n}(\mathbf{x}_\mu)\psi_{\mu n}^*(\mathbf{x}'_\mu) = \delta(\mathbf{x}_\mu - \mathbf{x}'_\mu) - \psi_{\mu 2S}(\mathbf{x}_\mu)\psi_{\mu 2S}^*(\mathbf{x}'_\mu). \tag{17}$$

In this case, the following integral in (15) was initially calculated:

$$I = \int d\mathbf{y}_1 \frac{e^{-b_n|\mathbf{x}_1 - \mathbf{y}_1|}}{|\mathbf{x}_1 - \mathbf{y}_1||\mathbf{y}_2 - \mathbf{y}_1|}\psi_{e1S}(\mathbf{y}_1) = \psi_{e1S}(0)\int d\mathbf{y}_1 \frac{e^{-b_n y_1}e^{-W_e|\mathbf{y}_1 + \mathbf{x}_1|}}{y_1|\mathbf{y}_1 + \mathbf{x}_1 - \mathbf{y}_2|} \approx \tag{18}$$

$$\psi_{e1S}(0)e^{-W_e x_1}\frac{4\pi}{b_n^2}\frac{(1 - e^{-b_n|\mathbf{x}_1 - \mathbf{y}_2|})}{|\mathbf{x}_1 - \mathbf{y}_2|} \approx \psi_{e1S}(0)e^{-W_e x_1}4\pi\left[\frac{1}{b_n} - \frac{|\mathbf{x}_1 - \mathbf{y}_2|}{2} + \frac{b_n|\mathbf{x}_1 - \mathbf{y}_2|^2}{6} + \cdots\right],$$

where an expansion in terms of the recoil parameter $a_1$ was also used. The contribution of the first term in the square brackets was equal to 0 due to the orthogonality of the muon wave functions, and the second term indicated the contribution of the leading order in $a_1$. Using the completeness condition, this contribution could be expressed in terms of integrals, which were calculated analytically. One of them took the following form:

$$J = \int d\mathbf{x}_1 d\mathbf{x}_2 d\mathbf{y}_2 |\psi_{\mu\,2S}(\mathbf{y}_2)|^2 \frac{|\mathbf{x}_1 - \mathbf{y}_2|}{|\mathbf{x}_1 - \mathbf{x}_2|}|\psi_{\mu\,2S}(\mathbf{x}_2)|^2|\psi_{e\,1S}(\mathbf{x}_1)|^2 = 1 + 28a_1^2 - \frac{20073}{64}a_1^3 + \cdots. \tag{19}$$

In the case of the $2S$ state, the contribution of the second term from the right side of (18) was determined by the following expansion:

$$\Delta E_{2S}^{(2)}(n \neq 2S) = M_e\alpha^2 a_1^2\left(-7 + \frac{20073}{256}a_1 + \frac{137165}{256}a_1^2\cdots\right). \tag{20}$$

If the muon was in the $2P$ state, then the analogous contribution was equal to

$$\Delta E_{2P}^{(2)}(n \neq 2P) = M_e\alpha^2 a_1^2\left(-5 + \frac{12441}{256}a_1 + \frac{76997}{256}a_1^2\cdots\right). \tag{21}$$

The total contribution of the first and second orders of PT in $\Delta H$ of the muon Lamb shift $(2P - 2S)$ had the form:

$$\Delta E(2P - 2S) = M_e\alpha^2 a_1^2\left[2 + 8(Z - 1) + a_1(\frac{151}{16} - 80(Z - 1)) + \right. \tag{22}$$

$$\left. a_1^2(-\frac{42997}{96} - 384\ln(4a_1) + 480(Z - 1))\right].$$

To improve the calculation accuracy, we considered the contribution of the third term in expansion (18). After the integration over the electron coordinate and expansion in $a_1$, we presented this contribution, as follows:

$$\Delta E_{2S}^{(2)}(n \neq 2S) = -\frac{M_e\alpha^2}{3}|\psi_{e\,1S}(0)|^2\frac{2\pi}{W_e^4}\int \psi_{\mu\,2S}(\mathbf{x}_2)d\mathbf{x}_2\psi_{\mu\,2S}(\mathbf{y}_2)d\mathbf{y}_2\times \tag{23}$$

$$\sum_n b_n \psi_{\mu\,n}(\mathbf{x}_2)\psi_{\mu\,n}(\mathbf{y}_2)\left[-\frac{1}{3}(\mathbf{x}_2\mathbf{y}_2)a_1^2 - \frac{1}{3}x_2^2 y_2^2 a_1^4 + \dots\right].$$

The first term in the square brackets yielded the leading order contribution in $a_1$. It was determined by the following square of the matrix element:

$$\Delta E_{2S}^{(2)}(2S \to nP) = \frac{M_e\alpha^2}{9}\sqrt{\frac{M_e}{M_\mu}}W_e W_\mu \sum_n \sqrt{\frac{n^2-4}{n^2}}||\int \psi_{\mu\,2S}(\mathbf{x}_2)\mathbf{x}_2\psi_{\mu\,n}(\mathbf{x}_2)d\mathbf{x}_2||^2. \quad (24)$$

The contribution of the discrete and continuous spectrum for transitions $2S \to nP$ were represented separately as:

$$\Delta E_{2S}^{(2)}((2S \to nP)) = \frac{M_e\alpha^2}{9}\sqrt{\frac{M_e}{M_\mu}}\frac{W_e}{W_\mu}\left(S_d^{\frac{1}{2}}(2S \to nP) + S_c^{\frac{1}{2}}(2S \to nP)\right), \quad (25)$$

$$S_d^{\frac{1}{2}}(2S \to nP) = \sum_{n=3}^{\infty}\sqrt{n^2-4}\frac{2^{17}n^6(n^2-1)(n-2)^{2n-6}}{(n+2)^{2n+6}} = 9.83655\dots, \quad (26)$$

$$S_c^{\frac{1}{2}}(2S \to nP) = \int_0^{\infty} 2^6 \frac{k(k^2+1)dk}{(k^2+\frac{1}{4})^{\frac{11}{2}}(1-e^{-\frac{2\pi}{k}})}e^{-\frac{4}{k}arctg(2k)} = 3.12747\dots. \quad (27)$$

For the $2P$ state, it was necessary to take into account the transitions $2P \to nD$ and $2P \to nS$. The contribution from transition $2P \to nD$ was numerically dominant:

$$\Delta E_{2P}^{(2)}((2P \to nD)) = \frac{2M_e\alpha^2}{27}\sqrt{\frac{M_e}{M_\mu}}\frac{W_e}{W_\mu}\left(S_d^{\frac{1}{2}}(2P \to nD) + S_c^{\frac{1}{2}}(2P \to nD)\right), \quad (28)$$

$$S_d^{\frac{1}{2}}(2P \to nD) = \sum_{n=3}^{\infty}\sqrt{n^2-4}\frac{2^{19}n^8(n^2-1)(n-2)^{2n-7}}{(n+2)^{2n+7}} = 21.44214\dots, \quad (29)$$

$$S_c^{\frac{1}{2}}(2P \to nD) = \frac{2^6}{3}\int_0^{\infty}\frac{k(k^2+1)dk}{(k^2+\frac{1}{4})^{13/2}(1-e^{-\frac{2\pi}{k}})}e^{-\frac{4}{k}arctg(2k)} = 2.71117\dots. \quad (30)$$

The contribution of the $2P \to nS$ transition was much smaller, but it was also important for achieving a good calculation accuracy. The structure of the expressions by which it was defined was the same as for other transitions:

$$\Delta E_{2P}^{(2)}((2P \to nS)) = \frac{M_e\alpha^2}{27}\sqrt{\frac{M_e}{M_\mu}}\frac{W_e}{W_\mu}\left(S_d^{\frac{1}{2}}(2P \to nS) + S_c^{\frac{1}{2}}(2P \to nS)\right), \quad (31)$$

$$S_d^{\frac{1}{2}}(2P \to nS) = \sum_{n=3}^{\infty}\sqrt{n^2-4}\frac{2^{15}n^8(n^2-1)(n-2)^{2n-6}}{3(n+2)^{2n+6}} = 0.90371\dots, \quad (32)$$

$$S_c^{\frac{1}{2}}(2P \to nS) = \frac{2^4}{3}\int_0^{\infty}\frac{k(k^2+1)dk}{(k^2+\frac{1}{4})^{11/2}(1-e^{-\frac{2\pi}{k}})}e^{-\frac{4}{k}arctg(2k)} = 0.22078\dots. \quad (33)$$

Along with the correction for the Coulomb interaction of particles $\Delta H$ in (3), there was another perturbation operator $\Delta H_{rec}$ on the nucleus recoil.

### 2.2. Nuclear Recoil Corrections

To date, we considered corrections that were determined by the $M_e/M_\mu$ mass ratio and could be called recoil corrections. At the same time, there were other corrections for the recoil of the nucleus with the mass $m_N$ of a three-particle system. They were determined

by the term of the Hamiltonian $\Delta H_{rec}$. In the first order of the perturbation theory in $\Delta H_{rec}$, the correction to energy levels was zero:

$$\Delta E_{rec}^{(1)}(2S, 2P) = \langle \psi_{2S,2P} | \Delta H_{rec} | \psi_{2S,2P} \rangle = 0. \tag{34}$$

In the second order of the perturbation theory, when the second perturbation operator was equal to $\Delta H$, in general, the correction for the level $2S$ was equal to

$$\Delta E_{rec}^{(2)}(2S) = \int \psi_{\mu 2S}(\mathbf{x}_\mu) \psi_{e1S}(\mathbf{x}_e) \left( \frac{\alpha}{x_{\mu e}} - \frac{\alpha}{x_e} \right) d\mathbf{x}_e d\mathbf{x}'_e d\mathbf{x}_\mu d\mathbf{x}'_\mu \tag{35}$$

$$\sum_{n,n'} \frac{\psi_{\mu n}(\mathbf{x}_\mu) \psi_{\mu n}^*(\mathbf{x}'_\mu) \psi_{en'}(\mathbf{x}_e) \psi_{en'}^*(\mathbf{x}'_e)}{E_{e1S} - E_{en'} + E_{\mu 2S} - E_{\mu n}} \frac{(-1)}{m_n} \nabla'_\mu \nabla'_e \psi_{\mu 2S}(\mathbf{x}'_\mu) \psi_{e1S}(\mathbf{x}'_e).$$

Here again, it was convenient for our calculation to isolate the muon state $2S$ from the total sum first. However, then the resulting integral vanished when integrated over angles: $\int \psi_{\mu 2S}(\mathbf{x}'_\mu) \nabla'_\mu \psi_{\mu 2S}(\mathbf{x}'_\mu) d\mathbf{x}'_\mu = 0$. If the muon was in the intermediate state $n \neq 2S$, then the corresponding contribution could be reduced to the form:

$$\Delta E_{2S,rec}^{(2)}(n \neq 2S) = \int \psi_{\mu 2S}(\mathbf{x}_\mu) \psi_{e1S}(\mathbf{x}_e) \frac{\alpha}{x_{\mu e}} \psi_{\mu 2S}(\mathbf{x}'_\mu) d\mathbf{x}_e d\mathbf{x}'_e d\mathbf{x}_\mu d\mathbf{x}'_\mu \tag{36}$$

$$\times \sum_{n \neq 2S} \psi_{\mu n}(\mathbf{x}_\mu) \psi_{\mu n}^*(\mathbf{x}'_\mu) \left( -\frac{M_e}{2\pi} \right) \frac{e^{-b|\mathbf{x}_e - \mathbf{x}'_e|}}{|\mathbf{x}_e - \mathbf{x}'_e|} \frac{(-1)}{m_N} (\nabla'_\mu \nabla'_e) \psi_{\mu 2S}(\mathbf{x}'_\mu) \psi_{e1S}(\mathbf{x}'_e).$$

The integral over the coordinate $\mathbf{x}'_e$ was calculated in (18). Taking the second term in square brackets in (18) and calculating the derivatives with respect to the coordinates of the wave functions, we rewrote the intermediate result for the correction in the following form:

$$\Delta E_{rec}^{(2)}(2S) = -\frac{\alpha M_e}{m_N} \frac{W^{5/2}}{\sqrt{2\pi}} \frac{W_e^4}{\pi} \int \psi_{\mu 2S}(\mathbf{x}_\mu) d\mathbf{x}_\mu |\mathbf{x}_\mu - \mathbf{x}'_e| \left[ \delta(\mathbf{x}_\mu - \mathbf{x}'_\mu) - \psi_{\mu 2S}(\mathbf{x}_\mu) \psi_{\mu 2S}(\mathbf{x}'_\mu) \right] \times \tag{37}$$

$$\frac{(\mathbf{x}'_e \mathbf{x}'_\mu)}{x'_e x'_\mu} d\mathbf{x}'_\mu d\mathbf{x}'_e e^{-W_e x'_e} e^{-\frac{1}{2} W_\mu x'_\mu} \left( 1 - \frac{1}{4} W_\mu x_\mu \right)$$

After that, the direct calculation of integrals over coordinates in two terms in square brackets (37) yielded the following result:

$$\Delta E_{rec}^{(2)}(2S) = -\frac{4\alpha M_e W_e}{m_N} \frac{(2 + 10a_1 + 6a_1^2 + 5a_1^3 + a_1^4)}{(1 + a_1)^5}. \tag{38}$$

In the case of the $2P$ state, the second-order contribution of PT was calculated in a similar way. The final analytical result for the correction to the $2P$ level was equal to

$$\Delta E_{rec}^{(2)}(2P) = -\frac{4\alpha M_e W_e}{m_N} \frac{(2 + 10a_1 + 10a_1^2 + 5a_1^3 + a_1^4)}{(1 + a_1)^5}. \tag{39}$$

From expressions (38) and (39), it follows that the value of the correction in the Lamb shift contains a small parameter $a_1^2$:

$$\Delta E_{rec}^{(2)}(2P - 2S) = -\frac{16\alpha M_e W_e}{m_N} \frac{a_1^2}{(1 + a_1)^5}. \tag{40}$$

In addition to the corrections for the Coulomb interaction (22), other corrections for vacuum polarization, nuclear structure, and recoil must also be taken into account in order to obtain the total value of the muon Lamb shift with high accuracy. The corrections of this type had been calculated previously in [23,24].

### 3. Variational Method

The analytical results obtained in the previous sections were verified using the variational method [11,13]. The trial wave function for a system of three particles in the S-state was presented in the following form:

$$\Psi(\boldsymbol{\rho}, \boldsymbol{\lambda}, A_{ij}^i) = \sum_{i=1}^{K} C_i \psi_i(\boldsymbol{\rho}, \boldsymbol{\lambda}, A_{ij}) = \sum_{i=1}^{K} C_i e^{-\frac{1}{2}[A_{11}^i \rho^2 + 2A_{12}^i \rho\lambda + A_{22}^i \lambda^2]}, \tag{41}$$

where the $C_i$ variables are linear variational parameters. The Jacobi coordinates $\rho$ and $\lambda$ were related to the radius vectors of the nucleus $\mathbf{r}_1$, muon $\mathbf{r}_2$, and electron $\mathbf{r}_3$:

$$\boldsymbol{\rho} = \mathbf{r}_2 - \mathbf{r}_1, \quad \boldsymbol{\lambda} = \mathbf{r}_3 - \frac{m_1 \mathbf{r}_1 + m_2 \mathbf{r}_2}{m_1 + m_2}, \tag{42}$$

where $A_{ij}$ is the matrix of nonlinear parameters. The problem was to find such values of the parameters and expansion coefficients that the average value of the Hamiltonian was minimal. To find the energies of bound states, the Schrodinger equation with the Coulomb interaction of three particles was reduced to solving a matrix eigenvalue problem of the following form:

$$HC = E^\lambda BC, \tag{43}$$

where the matrix elements $H_{ij} = < \psi_i | H | \psi_j >$ and $B_{ij} = < \psi_i | \psi_j >$ are calculated analytically using the variational wave functions, and $E^\lambda$ is one of the energy eigenvalues. The upper bound for the state energy of a system of three particles in the variational approach was provided by the smallest eigenvalue of the generalized eigenvalue problem.

The numerically obtained wave functions of various states made it possible to calculate various corrections to energy levels to increase the accuracy of the calculation. The radial distribution densities in $\rho$ and $\lambda$, as well as the root-mean-square values $\sqrt{< \rho^2 >}$, $\sqrt{< \lambda^2 >}$ were:

$$W(\rho) = \frac{8\sqrt{2}\pi^{5/2}}{< \Psi | \Psi >} \sum_{i,j=1}^{K} \frac{C_i C_j}{B_{22}^{3/2}} \rho^2 e^{-\frac{1}{2}\rho^2 \frac{detB}{B_{22}}}, \quad W(\lambda) = \frac{8\sqrt{2}\pi^{5/2}}{< \Psi | \Psi >} \sum_{i,j=1}^{K} \frac{C_i C_j}{B_{11}^{3/2}} \lambda^2 e^{-\frac{1}{2}\rho^2 \frac{detB}{B_{11}}}, \tag{44}$$

$$W(\rho, \lambda) = \frac{16\pi^2}{< \Psi | \Psi >} \sum_{i,j=1}^{K} \frac{C_i C_j}{B_{12}} \rho\lambda e^{-\frac{1}{2}[\rho^2 B_{11} + \lambda^2 B_{22}]} sh(B_{12}\rho\lambda), \quad B_{lk} = A_{lk}^i + A_{lk}^j, \tag{45}$$

$$< \rho^2 > = \frac{24\pi^3}{< \Psi | \Psi >} \sum_{i,j=1}^{K} C_i C_j \frac{B_{22}}{(detB)^{5/2}}, \quad < \lambda^2 > = \frac{24\pi^3}{< \Psi | \Psi >} \sum_{i,j=1}^{K} C_i C_j \frac{B_{11}}{(detB)^{5/2}}. \tag{46}$$

The radial distribution densities are shown in Figures 1 and 2 for muonic helium-3. These plots showed the presence of two characteristic distances in the particle system $He - \mu - e$. From (46), it followed that the root-mean-square values $\rho_N = \sqrt{< \rho^2 >}$ and $\lambda_N = \sqrt{< \lambda^2 >}$ for muon-electron helium ($He - 3, 4$) had close values: $\rho_{He} = 850 \ fm$, $\lambda_{He} = 91.3 \times 10^3 \ fm$.

Our numerical calculations of the variational method were carried out in a MATLAB environment [11]. Along with the Gaussian basis, the exponential basis was also used, as in [13]. The comparison of the numerical values of the energy of the states $2S^{(\mu)}1S^{(e)}$ and $2P^{(\mu)}1S^{(e)}$, which were obtained using these two bases, is presented in Table 1.

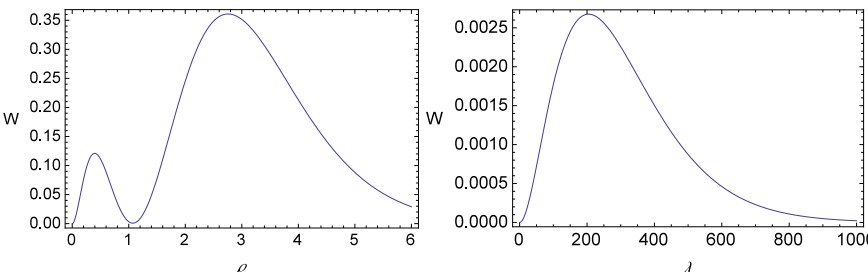

**Figure 1.** The radial distribution densities $W(\rho)$ and $W(\lambda)$ for the state $2S^{(\mu)}1S^{(e)}$. The variable values $\rho$ and $\lambda$ are presented in muonic atomic units.

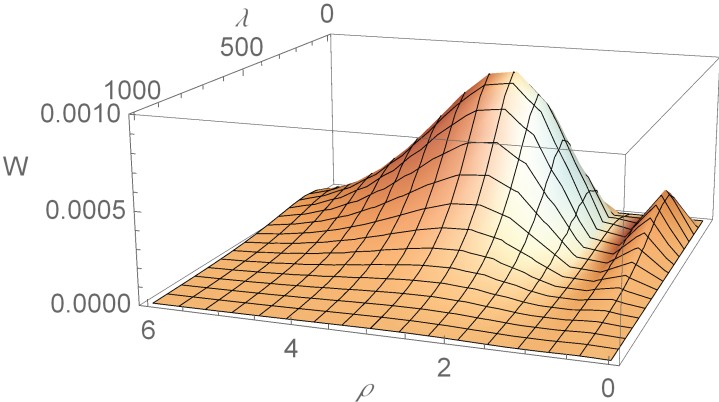

**Figure 2.** The radial distribution density $W(\rho, \lambda)$ for $(\mu e^3 He)$. The variable values $\rho$ and $\lambda$ are presented in muonic atomic units.

**Table 1.** Numerical values of the Coulomb corrections of the muon Lamb shift. The abbreviations G and Exp. denote Gaussian and exponential trial wave functions. The last column presents the results of the analytical calculation of the Lamb shift.

| $N - \mu - e$ | Basis | $2S$, $\mu$.a.u. | $2P$, $\mu$.a.u. | $(2P - 2S)$ | $(2P - 2S)$ |
|---|---|---|---|---|---|
| $^3_2He - \mu - e$ | G | $-0.48428998695$ | $-0.48428934525$ | 3.61 meV | 3.37 meV |
| | Exp. | $-0.48429003559$ | $-0.48428946173$ | 3.23 meV | |
| $^4_2He - \mu - e$ | G | $-0.48863657185$ | $-0.48863598056$ | 3.33 meV | 3.32 meV |
| | Exp. | $-0.48863665520$ | $-0.48863609292$ | 3.16 meV | |

## 4. Conclusions

When calculating some of the contributions analytically, we replaced the exact Green's function of the electron with the free one. In this case, we neglected corrections of the order of $\sqrt{M_e/M_\mu}$ in comparison with the calculated terms (see Equation (15)). Numerically, $\sqrt{M_e/M_\mu} \approx 0.1$, so the main error of analytical calculations was approximately 10 percent. By including the calculation of the Lamb shift in the framework of the variational method, we then had to verify the results of the analytical calculations, on the one hand, and on the other hand, improve the accuracy of the calculations. The variational method provides a very high accuracy in the calculation of energy levels. In Table 1, we defined the results with an accuracy of two digits after the decimal point in order to achieve a comparison with analytical results. The comparison of the results obtained by different methods has shown that these results agreed with each other within the limits of a possible theoretical error in analytical calculations. The slight difference in the variational results when using the exponential and Gaussian bases could be explained by the size of the bases and by the fact that the convergence with the Gaussian basis was slower than with the exponential wave functions. The maximum size of a basis with the Gaussian wave functions was approximately 400, and with exponential wave functions, approximately 2000.

The measurement of the Lamb shift in two-particle muonic atoms and ions by the CREMA [1] collaboration had made it possible to obtain an order of magnitude of more accurate values of the charge radii of the proton, deuteron, and alpha particles. This was accomplished after the precise calculations of the Lamb shift, which had included the effects of vacuum polarization, nuclear structure, relativistic corrections, and higher-order mixed-type effects in the $\alpha$ and particle mass ratio [23,24]. In this paper, we studied the influence of the particle coupling effects on the magnitude of the muon Lamb shift in three-particle muon-electron-nucleus systems. The presence of an electron led to an additional Coulomb interaction with the muon and nucleus, and it changed the magnitude of the Lamb shift, as compared to two-particle systems. The numerical values of the Coulomb correction for various muon-electron systems were as follows:

$$\Delta E(\mu e_3^7 Li) = 9.12 \ meV, \ \Delta E(\mu e_4^9 Be) = 14.27 \ meV, \ \Delta E(\mu e_5^{11} B) = 19.33 \ meV. \tag{47}$$

Considering (47) and the results of [23,24], we obtained the values of the muon $(2P - 2S)$ Lamb shift (Ls) (values were given for nuclei Li, Be, and B with the same spin 3/2):

$$E^{Ls}(_2^3 He) = 1263.23 \ meV, \quad E^{Ls}(_2^4 He) = 1382.43 \ meV, \tag{48}$$

$$E^{Ls}(Li) = 1540.90 \ meV, \ E^{Ls}(Be) = -1228.55 \ meV, \ E^{Ls}(B) = -7981.00 \ meV.$$

The Coulomb interaction of the particles in three-particle systems led to a small but significant change in the magnitude of the muon Lamb shift, as compared to two-particle muon systems. Accounting for the correction (47) was necessary to extract the value of the charge radius of the helion and the $\alpha$ particles with an accuracy exceeding 0.01 fm.

**Author Contributions:** The authors contributed equally to this work. All authors have read and agreed to the published version of the manuscript.

**Funding:** This research is supported by the Foundation for the Development of Theoretical Physics and Mathematics "Basis" (grant no. 22-1-1-23-1) and by the Russian Science Foundation (grant No. RSF 23-22-00143).

**Informed Consent Statement:** Informed consent was obtained from all subjects involved in the study.

**Data Availability Statement:** Not applicable.

**Acknowledgments:** This work is supported by the Foundation for the Development of Theoretical Physics and Mathematics "Basis" (grant no. 22-1-1-23-1) and by the Russian Science Foundation (grant No. RSF 23-22-00143).

**Conflicts of Interest:** The authors declare no conflict of interest

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
