# Peer review of "Three Particle Muon-Electron Bound Systems in Quantum Electrodynamics"

_atoms, doi:10.3390/atoms11020025_

Round 1

Reviewer 1 Report

 This an interesting and well written paper that calculates the
 muonic 2P-2S Lamb shift in muon-electron atoms and ions of helium,
 lithium, beryllium, and boron. The calculations involve a great
 deal of analytical work in which the fine structure constant and
 the electron-muon mass ratio are treated a perturbation
 parameters.  At the end of the paper, the results are compared
 with variational calculations involving Gaussian and exponential
 basis sets.

 My main criticism of the paper is just that there should be more
 detail given about the variational calculations presented in Table
 1.  Since the results differ by as much as 10%, it would be good
 to know which is right and which is wrong.  I would suggest the
 following:
 1.  give uncertainty estimates for both the perturbation
 calculations and the variational calculations.
 2.  say something about the size of the variational basis sets, how
 they were constructed and the rate of convergence with size.  What
 limited the accuracy to the small number of figures shown?

 As minor changes, I noted the following:
 1.  p. 1, line 25: change "more exact" to "more accurate" or "more
 precise."  A result is either exact or it is not.
 2.  p. 1, line 32:  why not use the standard notation keV in place of
 kiloelectronvolts.

 With these changes, I recommend the paper for publication.

Author Response

Dear Referee 1,

We send you our reply and revised version of the manuscript,

Sincerely yours, Alexei Martynenko

Reviewer 2 Report

See attached.

Author Response

Dear Referee 2,

We send you our reply and revised version of the manuscript,

Sincerely yours, Alexei Martynenko
